# Fast temporal decoding from large-scale neural recordings in monkey visual cortex

**Jerome Hadorn**[1][*], **Zuowen Wang**[1][*], **Bodo Rueckauer**[2],
**Xing Chen**[3], **Pieter R. Roelfsema**[4], **Shih-Chii Liu**[1]
[1] Sensors Group, Institute of Neuroinformatics, University of Zurich and ETH Zurich
[2] Radboud University Nijmegen
[3] University of Pittsburgh School of Medicine
[4] Netherlands Institute for Neuroscience

## Abstract

With new developments in electrode and nanoscale technology, a large-scale multi-electrode cortical neural prosthesis with thousands of stimulation and recording electrodes is becoming viable. Such a system will be useful as both a neuroscience tool and a neuroprosthesis. In the context of a visual neuroprosthesis, a rudimentary form of vision can be presented to the visually impaired by stimulating the electrodes to induce phosphene patterns. Additional feedback in a closed-loop system can be provided by rapid decoding of recorded responses from relevant brain areas. This work looks at temporal decoding results from a dataset of 1024 electrode recordings collected from the V1 and V4 areas of a primate performing a visual discrimination task. By applying deep learning models, the peak decoding accuracy from the V1 data can be obtained by a moving time window of 150 ms across the 800 ms phase of stimulus presentation. The peak accuracy from the V4 data is achieved at a larger latency and by using a larger moving time window of 300 ms. Decoding using a running window of 30 ms on the V1 data showed only a 4% drop in peak accuracy. We also determined the robustness of the decoder to electrode failure by choosing a subset of important electrodes using a previously reported algorithm for scaling the importance of inputs to a network. Results show that the accuracy of 91.1% from a network trained on the selected subset of 256 electrodes is close to the accuracy of 91.7% from using all 1024 electrodes.

## 1 Introduction

Recent advances in machine learning research have motivated researchers to better understand biological brains, as well as interfacing the brain with the environment. The authors in [19] demonstrate that hierarchical convolutional neural networks trained with goal-driven optimization result in neurally predictive models of the ventral visual cortex. A later work [20] showed that deep neural networks trained with an unsupervised procedure yields equal or better neural models in terms of the prediction of response patterns. Besides directly modeling the ventral visual system, another line of work focuses on decoding the neural activities [1, 3, 14, 17] with machine learning algorithms, in order to understand the properties of certain cortical area or achieve functionalities such as brain-to-text communication [17]. Decoding neural activity is an important area especially with the appearance of smaller and denser electrode arrays and the possible future of a large-scale cortical neuroprothesis, e.g, for the visually impaired population [7, 10, 2].

---

[*]Equal contribution. Correspondence to zuowen, shih@ini.uzh.ch

4th Workshop on Shared Visual Representations in Human and Machine Visual Intelligence (SVRHM) at the Neural Information Processing Systems (NeurIPS) conference 2022. New Orleans.

Various challenges appear for future large-scale recording / stimulation recording systems [6, 12, 5]. One challenge is the decoding latency of recorded responses to support a closed-loop neuroprosthesis system. Studies of this type, e.g [9], will enable further development of closed-loop strategies for dynamically changing the stimulation signals. A second challenge is to determine approaches that reduce computational resources for a portable system, e.g by decoding only a subset of electrodes useful for a particular task. In this work, we applied deep learning methods to determine if rapid decoding of recorded responses is possible within short time windows of tens to hundreds of milliseconds. The experiments are carried out on a dataset previously reported in [4]. The dataset consisted of recordings from 1024 electrodes implanted in both V1 and V4 areas of a macaque monkey which performed a visual discrimination task on a set of ten visual stimuli. We compare the latency to peak accuracy of the decoded responses on the recordings from V1 and V4 so we can determine if the accuracies from a higher-level or lower-level area would differ and if the latencies to peak accuracy would also be correspondingly higher. Code is available at `https://github.com/SensorsINI/V1V4-Neural-Decoding`.

## 2 Dataset and methods

This section describes the dataset and the decoding methods used in this study.

### 2.1 V1/V4 neural recording dataset

This dataset consists of recordings performed in experiments previously reported in [4]. In these experiments, two macaque monkeys were trained on a visual discrimination task on stimuli presented on a screen in front of the animal. The symbols consist of 8 familiar stimuli representing characters **I**, **U**, **A**, **L**, **T**, **V**, **S**, **Y**; a semi-familiar stimulus **J**, which the monkey has only seen limited times and one novel stimuli □, which the monkey has never been trained on. Each animal was implanted with 16 Utah electrode arrays in the V1/V4 areas of the visual cortex (see Figure 1, also in [4]). Each Utah array has 64 independent electrodes, making 1024 electrodes (channels) in total, where 896 electrodes are from V1 and 128 electrodes are from V4.

This study used only the recordings of one animal. In total, there were 2780 trials where the animal performed the discrimination task over the span of three days. Each trial lasts 1500 ms. After 300 ms of fixation (pre-stimulus phase), one of the ten stimuli is presented on the screen. The stimulus presentation lasts 800 ms (stimulus phase). After the remaining 400 ms (post-stimulus phase), the monkey makes its decision about the stimulus identity. The preprocessing of the recordings is described in [4]. The average response, i.e. the multi-unit activity (MUA), of one V1 electrode across all trials is shown in Figure 1 (right) where one can see clear response peaks a short time after both onset and offset of the stimulus presentation.

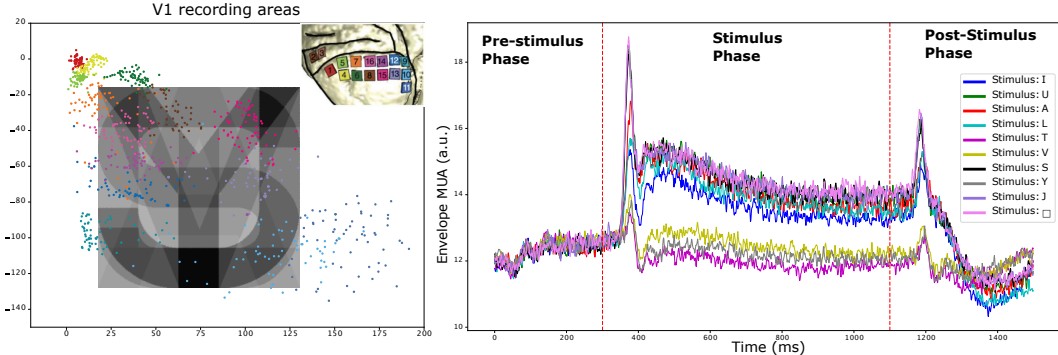

Figure 1: Left: Location of V1 electrodes relative to the stimulus location in the visual field. Inset of figure shows the implant locations of the 16 recording arrays and is adapted from [4]. Right: Averaged responses across trials from an example V1 electrode (541) for each of the 10 stimulus classes.

## 2.2 Neural decoding

We describe next the two classes of decoding methods used on the recordings. We first conduct dimension reduction and time warping on the time series data before visualization. We also present here supervised decoding methods used in the study, and the squeeze and excitation module for determining channel importance.

**Dimension reduction and visualization** We first remove the pre-stimulus phase of the recordings. Following [17], we first perform PCA to reduce the dimension of each time series recording from V1/V4/V1+V4. The corresponding dimensions of 892/128/1024 are reduced to 15. Then we align the neural recordings by using shift warping [13, 18] so as to reduce trial-to-trial temporal differences. Finally, we apply the t-distributed Stochastic Neighbor Embedding (t-SNE) [16] method to reduce the dimensions down to 2.

**Supervised neural decoding** We apply two deep learning models to the data: CNN and GRU. The CNN architecture has three 1-D convolutional layers, each followed by the ReLU activation and batch normalization. Each convolutional layer has kernel size 3 and outputs 32 channels. The GRU network has 4 bidirectional layers, each with hidden size 256. When using all 1024 input electrodes, the CNN and GRU models use 105k and 1482k trainable parameters respectively. The inputs to the decoding model are multi-dimensional time series. The dimension equals to the cardinality of the subset of included electrodes and each time step corresponds to 1 ms of neural recording data.

**Squeeze and excitation network** To determine the channel importance, we map the input $X \in \mathbb{R}^{C \times T}$ ($C = 1024, T = 1500$) to the network through a 1-D version of the 'Squeeze-and-Excitation (SE)' block [8] before the first layer of our CNN model (see Appendix A.1). We first use adaptive average pooling on each channel, resulting in $X' \in \mathbb{R}^{1 \times C}$ and then use a two-layer fully-connected block $F_{ex}(\cdot, W)$ on the squeezed (flattened) channels to obtain the scaling factors $s \in \mathbb{R}^{1 \times C}$. The fully connected block uses a sigmoid function as the activation function on the output layer so it always outputs values in $(0, 1)$. The final input $\tilde{X}$ to the CNN model is the channel-wise $X$ reweighted by these scaling factors, namely $\tilde{X} = s \cdot X$.

**Training** For all models, we use the Adam optimizer with an initial learning rate of 0.001. For the GRU and CNN, we used a dropout rate of 0.6 and 0.5 respectively. The test results are obtained by 10-fold cross validation for all experiments except for the results in Figure 3 which are based on a 5-fold cross validation. All experiments are conducted on an Intel(R) Xeon(R) W-2195 CPU and one Nvidia RTX 2080ti GPU and every run takes < 1 hour. Dataset sizes and number of validation splits are outlined in Appendix A.3.

## 3 Experiments

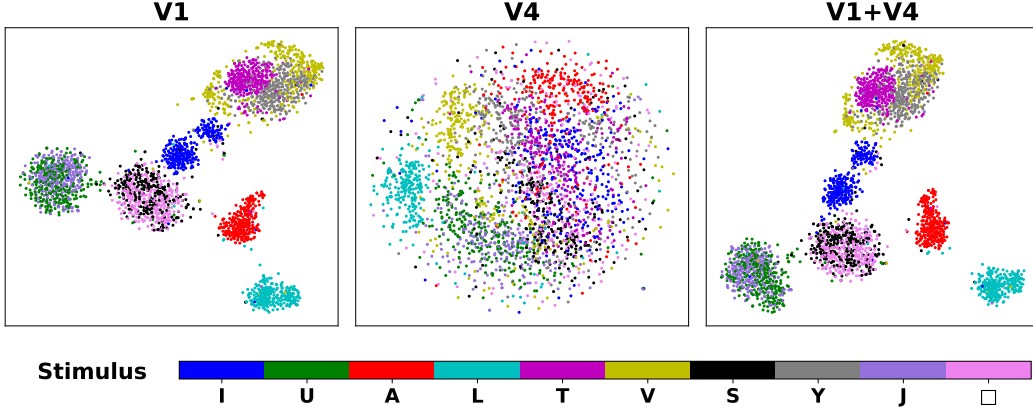

Figure 2: t-SNE visualization of the neural recordings from V1 (left), V4 (middle) and both V1 and V4 (right). For all three visualizations, every dot represents a dimension-reduced trial. Stimuli **U** and **J** share overlapping clusters, as well as stimuli □ and **S**. Stimuli **V**, **Y** and **T** are grouped closely.

## 3.1 Dimension reduction and visualization of V1 and V4 neural recordings

We first use the dimension reduction method in Section 2.2 before applying the unsupervised clustering method. The intent here is to determine the separability of the recorded data in response to the stimuli classes. Figure 2 shows the results for V1, V4, and both V1 and V4 recordings (described in 2.2) plotted in 2D. The stimuli clusters are more separated in the V1 recordings (left) compared to V4 (middle). When combining V1 and V4 data (right), there is no obvious increase in the separability of the clusters. In addition, we see that the clusters for stimuli pairs **U** and **J**, **S** and □ are overlapping in Figure 2 (left). The results show that we can already differentiate to some extent, the clusters corresponding to the individual stimuli.

## 3.2 Neural decoding with supervised deep learning

**Deep learning model results** We investigate the decoding accuracies on the recordings from V1 and V4 using two deep neural network architectures, namely a CNN and a GRU. We also compare the accuracies from using different response durations ([150, 300, 800, 1200] ms) after the visual stimulus onset at **300 ms**.

| | Duration after onset | **Mean cross validation accuracy $\pm$ standard deviation** (in %) | | | |
|---|---|---|---|---|---|
| | | 150 ms | 300 ms | 800 ms | 1200 ms |
| CNN | V1 | $87.80 \pm 1.52$ | $90.03 \pm 1.66$ | $90.36 \pm 1.64$ | $\mathbf{90.79 \pm 1.64}$ |
| | V4 | $48.77 \pm 2.27$ | $75.36 \pm 2.03$ | $\mathbf{81.55 \pm 2.07}$ | $81.04 \pm 1.84$ |
| | V1+V4 | $87.84 \pm 1.79$ | $90.14 \pm 1.69$ | $90.25 \pm 1.91$ | $\mathbf{91.40 \pm 1.78}$ |
| GRU | V1 | $85.89 \pm 1.60$ | $89.10 \pm 1.72$ | $89.53 \pm 1.67$ | $\mathbf{89.71 \pm 1.51}$ |
| | V4 | $57.55 \pm 5.51$ | $72.58 \pm 2.65$ | $\mathbf{83.09 \pm 2.29}$ | $80.86 \pm 1.56$ |
| | V1+V4 | $85.79 \pm 2.31$ | $89.60 \pm 1.57$ | $90.21 \pm 0.96$ | $\mathbf{91.11 \pm 1.43}$ |

Table 1: Mean and standard deviation of the cross validation accuracy from training CNN and GRU models on the data across different durations after stimulus onset.

As shown in Table 1, both CNN and GRU achieve $\sim 90\%$ accuracy on durations $\geq 300$ ms from either V1 or V1+V4 recordings. Lower accuracies were obtained with the V4 recordings across all durations. The lower accuracy from V4 matches the unsupervised decoding results in Section 3.1 which shows that the stimulus clusters overlap significantly. Interestingly, the decoding accuracy in V1 improves little as the duration increases (from 87.80% to 90.79% for CNN) while the decoding accuracy for V4 increases significantly when going from a duration of 150 ms to 300 ms. This could be explained by the additional latency for activity to propagate from V1 to V4 [15, 11]. The best decoding accuracy of 91.7% was achieved with a CNN model using the entire recording length of 1500ms from all 1024 electrodes. Because the CNN and GRU gave similar results, the remaining experiments are carried out with the CNN.

**Temporal characteristics and real-time decoding** To determine if decoding can be fast, we investigate the accuracies from using only data within a short time window instead of the whole recording. From this, we will also determine the answers to two additional questions: One, during the 1500 ms of a recording, which parts contain information which allows the highest decoding accuracy and is there a difference between the V1 and V4 recordings? Two, how does the decoding accuracy change in the two areas for different time windows.
In Figure 3 we show the decoding accuracy from V1 and V4 using a moving time window of either 20, 30, 150, or 300 ms across the whole recording and a stride of 20 ms. For V1 recordings, the curve quickly rises $\sim 60$ ms after stimulus onset while the V4 curves rise slower, $\sim 90$ ms after stimulus onset. In the case of V1, increasing the time window from 20 ms to 300 ms did not affect the temporal evolution of the accuracy values over the stimulus phase. While the peak accuracy from V1 slightly decreases for a time window of 300 ms compared to 20 ms, ($\sim 90\%$ to $\sim 84\%$), the peak accuracy of the V4 curves show a much larger drop when going between these two time windows ($\sim 80\%$ to $\sim 50\%$). These results suggest that V4 neurons have a larger temporal integration window and therefore the accuracy drops when using a smaller time window. The results also show a slight peak in decoding accuracy $\sim 100$ ms after stimulus offset for both V1 and V4.

**Electrode selection** The receptive fields of the different electrodes, particularly on the same array, can have a large degree of overlap. With this large-scale array matrix, it is highly probable that a subset of electrodes suffices for the task and the computational load on the decoding network would also be

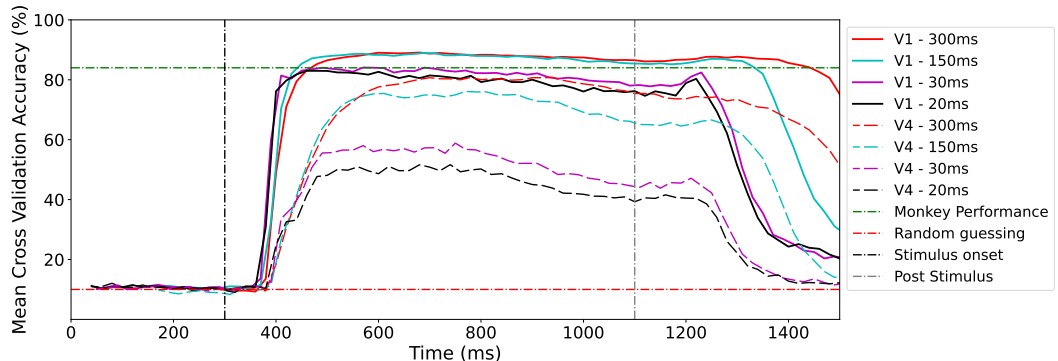

Figure 3: Mean accuracy of a CNN trained using one out of 4 moving time windows (20, 30, 150, and 300) ms and stride of 20 ms. The accuracy for random guesses is at 10% for the 10 stimuli.

reduced, benefiting a portable prosthesis. To determine the important electrodes for the task, we use the SE block described in Section 2.2. We chose the top-$k$ (important) electrodes from the ones with the highest scaling factor. We then retrained a CNN on these $k$ electrodes over the 1500ms recordings. The results in Figure 4 show that the mean accuracy of 91.1% with 256 electrodes is close to the mean accuracy of 91.7% using all 1024 electrodes. By decreasing the number of electrodes following the importance scale, the accuracy drops slightly even with 64 electrodes and then decreases faster as the number of electrodes further decreases. The receptive fields for some cases are shown in Appendix A.4.

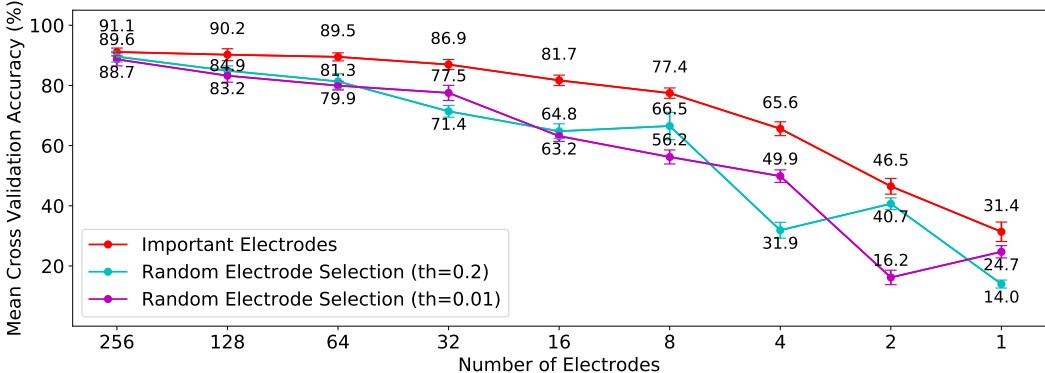

Figure 4: Mean cross validation accuracies from CNNs trained with reduced number of electrodes using the subset of important electrodes and "random electrodes". Error bars represent the standard deviation.

We perform the same experiment using the random selection of electrodes from a set of electrodes where their receptive field overlap with the stimulus location is above a threshold, $th$. For $th = 0.01$, we obtain 744 electrodes (see Appendix A.2 for details) while for $th = 0.2$, we obtain 450 electrodes. Random selection is then done on these sets. The results in Figure 4 show that the accuracy drops in a similar manner as the number of electrodes decrease but the accuracies for both random sets are lower than those obtained from the electrodes selected using importance.

## 4 Related work

A few papers have reported results from the decoding of a reasonably large number of electrodes. For example, the recordings from 192 electrodes implanted in the human motor cortex, allowed [17] to decode imagined handwriting movement activity with a recurrent neural network decoder. The decoding can be completed within 0.3-1 s. In [1], eight Neuropixel probes, each capable of recording from up to 384 electrodes, were placed across multiple brain areas. The recordings were

used to decode the behaviour of mice but the decoding latency was not reported. The authors in [9] studied the object solution time (OST) of a linear decoder trained on an object recognition task using recordings from 424 electrodes in the primate IT neurons. They showed that the OST can vary between 115 ms to 145 ms depending on the difficulty of the images.

## 5    Discussion

This work addresses two foreseen challenges in future large-scale recording / stimulation systems [12] possibly for a neuroprosthesis [2]: 1) their real-time decoding capability 2) the selection of a subset of electrodes to reduce the system resource requirements for a portable system. The results in this work show that real-time decoding from all 1024 electrodes gives an accuracy of $\sim 84\%$ even for time windows of 150 ms thereby providing possible low-latency feed-back in a closed-loop system. In addition, the decoding accuracy of the V4 neurons require a larger time-window than that of V1 neurons. Our results also show that decoding is possible with as few as 256 electrodes. We tested the latency of running the CNN model on a Jetson Nano operating in the 5W mode. For a 150 ms time window, the network inference takes 9.88ms when all electrodes were used. The effectiveness of our decoding schema for recordings under more natural viewing conditions needs to be tested in the future, as well as its generalizability and robustness across different subjects.

## Acknowledgments and Disclosure of Funding

This project has received funding from the European Union's Horizon 2020 research and innovation programme under grant agreement No 899287.

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

# A Appendix

## A.1 Squeeze and excitation network

We adapt the SE block from work [8] to process 1D data and only use it for the first layer. The SE block mainly consists of two parts: (1) a squeeze net $F_{sq}(\cdot)$, which in our case is a adaptive average pooling which reduces $T$ timesteps to 1 with a learnable kernel. (2) an excitation net $F_{ex}(\cdot, W)$, in our case a fully connected network which can be represented as $F_{ex}(\cdot, W) = Sigmoid((W_2(ReLU(W_1(X')))))$ where $W_1 \in \mathbb{R}^{C|16 \times C}, W_2 \in \mathbb{R}^{C \times C|16}$. It takes the output of the squeeze net $X'$ as input and generates channel-wise scaling factors $s \in \mathbb{R}^{1 \times C}$. These scaling factors are then multiplied channel-wise with the original input $X$. Figure 5 shows an illustration of the adapted SE block we use in this work.

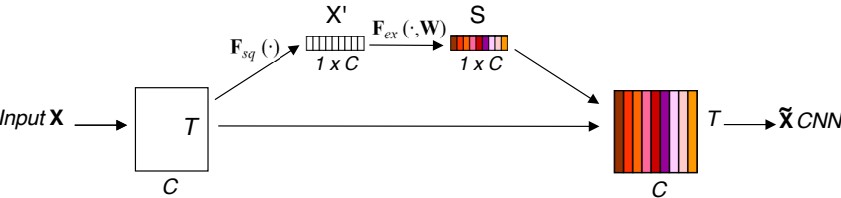

Figure 5: SE block on the input layer to determine the scaling factors of the channels. This figure is adapted from [8].

## A.2 Random electrode selection

Some electrodes were implanted further away from the location where the stimulus was placed. We filter out these electrodes before doing a random selection for the random electrode selection experiments in Figure. 4.

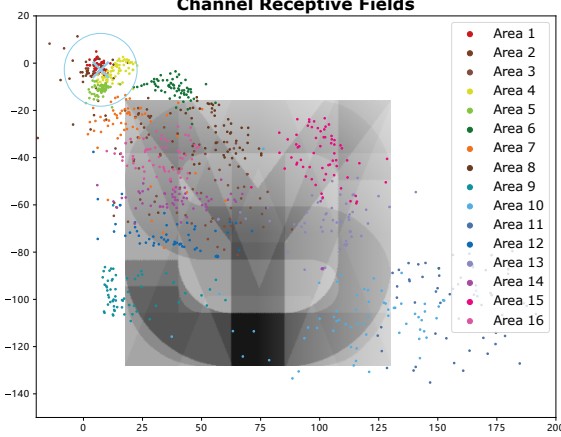

Figure 6: An example electrode whose receptive field (RF) does not overlap with the position where the visual stimuli is placed. The light blue cross marks the RF center of a specific electrode and the light blue circle marks the RF extent. Figure is adapted from [4].

## A.3 More training details

The results from the deep learning models as reported in Table 1 of Section 3.2, were generated over 10 splits of the original dataset of 2780 recordings. The training set, the validation set, and the test set contained 72%, 18% and 10% respectively of the recordings. These different splits were used in computing the importance weights which were then used to obtain the results from training CNNs with only $k$ electrodes as shown in Figure 4. Because of the long simulation time for getting the real-time decoding results in Figure 3 especially with short time windows, we did only 5 splits and

divided the original dataset so that the test set was 20% of the recordings in the original dataset, while the training set and validation set contained 64% and 16% respectively of the recordings.

## A.4  Choosing important electrodes

Figure 7 shows the top-k important electrodes determined by ranking the channel importance weights of the SE-module used in Figure 4.

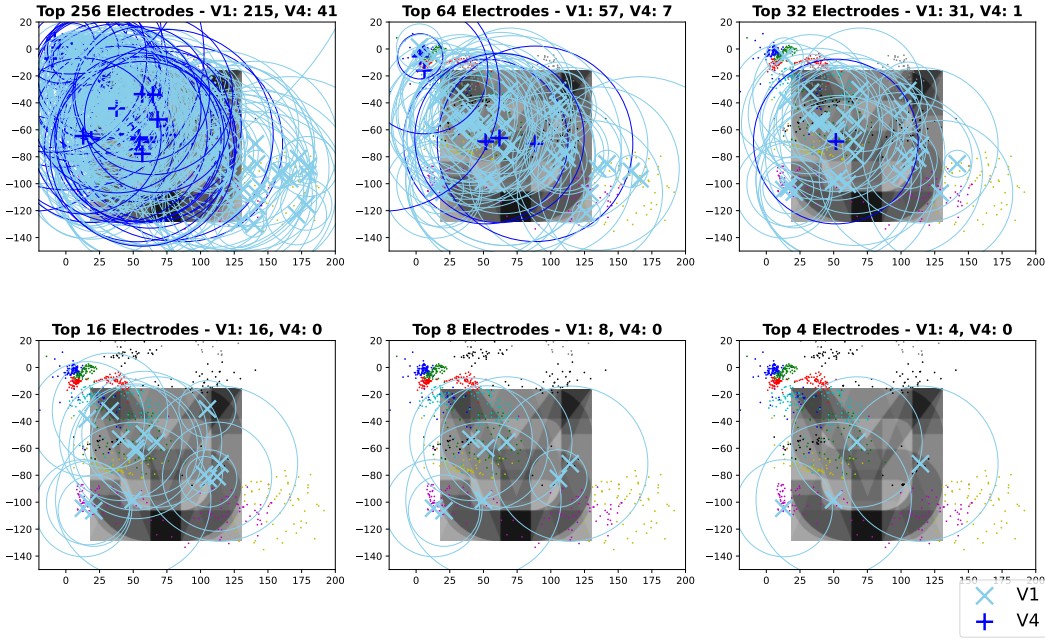

Figure 7: The receptive fields of the top-k most important electrodes corresponding to those with the largest SE scaling factors.

