# OpenReview forum: "Fast temporal decoding from large-scale neural recordings in monkey visual cortex"
_NeurIPS.cc/2022/Workshop/SVRHM — SVRHM Poster_

### Official Review · Reviewer_67KW · 2022-10-13
**a useful contribution**

**Rating:** 6
**Confidence:** 4

**Review:**


This submission evaluated the decoding performance of V1 and V4 populations (1024 electrodes recording) in a primate visual discrimination task using different time windows and different sample sizes (i.e., # of electrodes). They found using a short 30ms window only leads to a small drop in the decoding performance in V1, but not in V4.

I found these results interesting and relevant.

The authors motivate the work with real-time decoding. How long does it take to run the classifiers? Obviously, to do real-time decoding for a closed-loop system, the algorithm needs to be first enough.

---

> ### Author Response · Authors · 2022-12-28
> **Response to Reviewer 67KW**
>
> We thank the reviewer for their comments.
>
> The runtime latency of the decoding network when implemented on a Jetson Nano operating in the low power (5W) mode is around 9.88 ms when using a 150 ms input time window. The latency and power consumption can be further reduced by building custom integrated circuits for the final network.

---

### Official Review · Reviewer_bRQV · 2022-10-13
**A well written paper with some important issues**

**Rating:** 4
**Confidence:** 4

**Review:**

Overall, the paper is well written and is of high quality. The authors showed that decoding of electrophysiological recordings can be improved with the use of deep learning. The authors state that the purpose of this work is to improve closed-loop systems for neuroprostheses.

Pros:
Well written and clear
Thorough characterization of different neural networks
Important implications for medicine

Cons:
Missing some novelty, similar analyses have been previously performed with similar results, so it is unclear exactly what novel finding the paper presents.
Methodologically, clustering on the output of t-SNE is generally not recommended because of the propensity of t-SNE to create false clusters in the data, which could confound the results.

---

> ### Author Response · Authors · 2022-12-28
> **Response to Reviewer bRQV**
>
> We thank the reviewer for their comments.
>
> For clarification, we used t-SNE only for visualization of the data (similar to https://doi.org/10.1038/s41586-021-03506-2) We did not apply any unsupervised clustering algorithms for the reported accuracy results in this work. The t-SNE visualization is in line with the decoding results that recordings from V1 are easier to decode than from V4 for this task.

---

### Official Review · Reviewer_dRNf · 2022-10-14
**Clear decoding project on unique dataset with strong results**

**Rating:** 8
**Confidence:** 4

**Review:**

The project uses a unique monkey physiology dataset based on Utah array recordings, aimed at developing visual prostheses. It demonstrates and investigates DNN-based decoding of letter characters from early visual areas V1 and V4, presented during a visual discrimination task.

Overall this decoding project is quite straight-forward, the writing is clear, the methods are well motivated and described in detail, and I only see minor points where it could be improved. The analysis of the decoding performance shows strong results within the controlled domain of decoding a small number of characters. The t-SNE projection in particular is interesting to see.

A few points:

* The authors should briefly motivate the idea of a closed-loop neural prosthesis within the paper, and how this decoding project would support this. The connection between neural decoding and neural prostheses are not obvious to the reader.

* Relatedly, while the decoding performance seems high, it should be discussed whether it is sufficient for the use case (if there is a specific one).

* Can it be made explicit what the input and the output of decoding models are? It is implicit right now and is simple to add to make it more readable for people outside the field.

* Could the authors explicitly state the train-, validation and test set sizes (or splits)?

* The time windows seem quite long, which does erase a lot of potentially usable information. I understand the pragmatism behind it (given the good results), but am wondering whether the arsenal of modern neural networks doesn't have methods that can look at the signal in more detail.

* 120 typo: Two, how does ... for different time windows[?]

---

> ### Author Response · Authors · 2023-01-05
> **Response to Reviewer dRNf**
>
> We thank the reviewer for the detailed review. The replies below correspond to the order of the points in the review.
>
> 1. We have added a brief motivation in the paper about the usefulness of being able to decode in real-time the recordings of the electrodes.
> 2. We have now added more discussion points about the limitations of the results. More natural viewing conditions will need to be tested in future work similar to the use of the neural prosthesis in practical applications.
> 3. The input and output of the decoding model are now added to Section 2.2 under ‘supervised neural decoding’
> 4. The train/validation/test set description is now added to Appendix A.3.
> 5. We believe a 20ms time window is a short time for the chosen task, however, we agree with the reviewer that there could be more interesting information in the detailed signal.

---

### Official Review · Reviewer_Uk2C · 2022-10-14
**Interesting preliminary empirical work but some questions remain**

**Rating:** 7
**Confidence:** 4

**Review:**

This work attempts to look at the possibility of performing real-time decoding from large-scale neural recordings in visual cortex. Specifically, the authors record from V1 and V4 and try to decode letter identity by neural responses from a large-scale electrode array. The authors show that decoding these letters can be done with high accuracy using a basic CNN decoder from V1 responses and that this decoding is successful even using shortened time windows (for V1) suggesting the possibility of real-time decoding for neuroprostheses. However, they find that in V4 the decoding accuracy is worse and takes longer time windows. The most interesting study in my opinion- is the analysis of robustness to which electrodes get chosen and they show that surprisingly by using a simple scheme to identify "important electrodes" you can actually solve the task with much fewer electrodes (down to 64).

One first comment is that the authors try to make it seem like this is the first study to look at latency in decoding in visual cortex but they fail to cite "Evidence that recurrent circuits are critical to the ventral stream’s execution of core object recognition behavior" (Kar et al 2019) which is a pretty strong piece of work in this field, albeit looking at IT and natural object category decoding. Kar et al. 2019 do use much fewer total electrodes (96), but their results are arguably close to the direction that is proposed in the future work (looking at decoding with natural images) so it is worth citing and mentioning how this work relates.

Additionally, with the basic task of letter decoding, I think the results comparing V1/V4 are arguably unsurprising as it is natural that V4 decoding will require longer latencies (due to the latency of visual information making it through visual cortex), and it is also natural that V1 will provide better letter decoding since V1 neurons are much more sensitive to discriminating edges/lines. That being said, these results are a nice verification of these concepts and again performing these experiments with a large electrode setup does seem to be interesting and valuable.

As stated before, I think for the practical application of prostheses, the robustness study is quite interesting and valuable to the community. I agree with the authors that the true test of these ideas will be in extending to more complex images/decoding tasks, but they provide a solid foundation to build on. I am not familiar with all of the literature, but am taking the authors on their word that this is the first similar study to look at these topics of temporal decoding with the larger scale arrays that they are using.

As some points of clarification, I think it would also be worth describing the authors' thoughts on how well these decoders can generalize across different animals/subjects and if that will be an issue in making this practical. Also, again the authors should discuss how their results compare or are impacted by Kar et al.'s findings that recurrent models are necessary in IT for decoding of certain object categories. For example, in this experiment is the decoding of certain letters harder or easier than others (i.e. do certain letters require longer latencies than others?). Also do the authors believe these results will generalize to more stimuli?

In summary, I believe the work is interesting as a practical study and has lots of room to build on but there should be more discussion on the prior work (see above) and also more analysis or hypotheses about what the authors expect when trying to generalize to natural stimuli, more stimuli, across animals etc. (since these are all things that will ultimately determine if this method can work in a real neuroprosthetic setting).

---

> ### Author Response · Authors · 2023-01-05
> **Response to Reviewer Uk2C**
>
> Thanks for the detailed feedback and comments! We address the comments below:
>
> 1. Thanks for pointing us to the work of Kar et al, 2019 which uses a linear classifier to decode recordings from large-scale multi-electrode arrays implanted in the IT cortex.  As we understand it, our definition of latency is different. We use “latency” to refer to the minimal decoding time window size for achieving high accuracy. In Kar et al, “latency” is the extra time it takes the neural decoder to reach monkey performance when decoding challenging images compared to control images. The work of Kar et al is still very relevant to this work and we have added it to the paper.
> 2. In response to the reviewer’s question about whether these decoders can generalize across different animals/subjects, per-subject calibration or fine-tuning of the decoder will likely be required to compensate for patient-specific differences in both implant placement and the recordings. Recent advances in few-shot [http://dx.doi.org/10.1145/3386252] and continual learning [http://dx.doi.org/10.1016/j.neunet.2019.01.012] will enable training of the network for personal use and will be tackled in future research.
>
> We expect that the network and training scheme will need to be modified for natural viewing conditions. We hypothesize that models with a stateful component, such as recurrent (convolutional) neural networks, will be well suited to perform temporal integration over naturalistic visual sequences and act as noise and Kalman filters (similar to https://ieeexplore.ieee.org/document/8717140). This initial work allows us to determine if the recordings can be used to train a simple network for the classification task.